# The First 100 Days: Establishment and Effectiveness of Campus Protection Measures at a College during the COVID-19 Pandemic

**DOI:** 10.3390/healthcare8030308

**Published:** 2020-08-28

**Authors:** Tsan-Chang Chang, Mei-Ying Lin, Jui-Chi Huang, Cheng-Tung Yen, Ching-Hui Li, Woan-Ching Jan, Huei-Ying Huang, Chien-Liang Liu, Yu-Jen Chen

**Affiliations:** 1Department of Nursing, MacKay Junior College of Medicine, Nursing and Management, Taipei 112021, Taiwan; s354@mail.mkc.edu.tw (T.-C.C.); s121@mail.mkc.edu.tw (W.-C.J.); 2Department of Applied Foreign Languages, MacKay Junior College of Medicine, Nursing and Management, Taipei 112021, Taiwan; s450@mail.mkc.edu.tw; 3General Education Center, MacKay Junior College of Medicine, Nursing and Management, Taipei 112021, Taiwan; s375@mail.mkc.edu.tw (J.-C.H.); s104@mail.mkc.edu.tw (C.-T.Y.); 4Department of Cosmetic Applications and Management, MacKay Junior College of Medicine, Nursing and Management, Taipei 112021, Taiwan; s006@mail.mkc.edu.tw (C.-H.L.); s054@mail.mkc.edu.tw (H.-Y.H.); 5Department of Surgery, MacKay Memorial Hospital, Taipei 104217, Taiwan; 6Department of Medicine, MacKay Medical College, New Taipei City 252005, Taiwan; 7Department of Radiation Oncology, MacKay Memorial Hospital, Taipei 104217, Taiwan; 8Department of Medical Research, China Medical University Hospital, Taichung 404332, Taiwan; 9Department of Medical Research, MacKay Memorial Hospital, New Taipei City 251404, Taiwan

**Keywords:** campus protection measures, coronavirus, COVID-19, pandemic, college, effectiveness

## Abstract

To prevent transmission of the coronavirus, we established the campus protection measures for coronavirus disease 2019 (COVID-19) (CPMCV-19) and analyzed the effectiveness and cost in practice. This project was set in Taiwan. We organized an anti-epidemic task force team from multidisciplinary co-workers to establish the CPMCV-19. The essential components were as follows: no close contact communication, sterilization, temperature control, social distancing, activity restrictions, personal hygiene control, and situational awareness. During 100 days of operation, the mean time spent for frontal temperature measuring was 2.7 ± 0.3 s per person. The mean on-duty time for individual personnel to control the gate and measure temperature was 3.5 h per day. In total, 31 persons with loss of taste/smell or fever were detected on campus and sent to hospital for screening within 1 h. A total of 6 persons were instructed to observe self-health management due to possible contact or travel history, and none were diagnosed with COVID-19 infection. A total budget of USD 27,100 was used for CMPCV-19 in this period. The established campus protection measures for COVID-19 were practical and might be effective. They can be used as reference for schools in a pandemic, such as COVID-19.

## 1. Introduction

The coronavirus disease 2019 (COVID-19) is a respiratory disease caused by the novel coronavirus, also known as severe acute respiratory syndrome coronavirus 2 (SARS-CoV-2), which was first detected during an investigation of an outbreak in Wuhan, China [1,2,3,4,5]. This infectious disease has been spreading worldwide, and the World Health Organization declared a pandemic on 11 March 2020 [6]. As of June 2020, a total of 921 million confirmed cases and 476 thousand confirmed deaths in 216 countries/areas were noted and continue to increase.

The COVID-19 pandemic has affected entire societies, including educational systems, in countries all around the world [7,8,9]. Since the first confirmed case of COVID-19 in Taiwan was officially announced on 21 January 2020 [10,11], the educational system has been challenged to decide the critical cut-off for closure of face-to-face courses in schools. Given that schools share the same responsibilities as the rest of society and need to take precautions during the outbreak of diseases, efforts to prevent transmission of this coronavirus are critically important. Towards this end, we established and began execution of campus protection measures for COVID-19 (CPMCV-19) on 26 January 2020, 5 days after the announcement of the first confirmed case in Taiwan.

This study was conducted in MacKay Junior College of Medicine, Nursing and Management, Taiwan. This school comprises about 4200 students aged between 16 and 20 years old. In the present study, we analyzed the effectiveness and cost in practicing the CPMCV-19 to provide informative statistics, as well as practical information, for other schools facing the challenges during the COVID-19 pandemic.

## 2. Materials and Methods

### 2.1. Organization for Multidisciplinary Task Force Team Managing COVID-19 Related Issues

The selection of team members was mostly accomplished using existing personnel and operation designations within the original school structure. All department principal supervisors and selected secondary supervisors, 30 persons in total, were recruited and assigned to various task groups. The organization chart of this multidisciplinary task team is illustrated in Figure 1.

The investigation was carried out following the rules of the Declaration of Helsinki of 1975. All the 30 persons involved had been informed of this study and manuscript, and their signed consent was obtained.

### 2.2. Establishment of Campus Protection Measures for COVID-19 (CPMCV-19)

The task team members established the campus protection measures according to the guidelines from Taiwan Centers for Disease Control (CDC) with timely updates. The details were further customized by team members from each unit. All protection measures were adjusted through a rolling correction mechanism. Team discussions were conducted through mobile application LINETM (LINE Corporation, Tokyo, Japan), institutional e-mail system, teleconferences, or face-to-face meetings.

### 2.3. Decision-Making Algorithm in Managing Suspected Infection on Campus

Event simulation was conducted for the development of countermeasures including detected fever, reported symptoms related to COVID-19, confirmed case on campus, and others. In order to efficiently and appropriately cope with the situation, we set up a decision-making algorithm including criteria for a 14-day school closure when 2 confirmed cases are reported within a 14-day period, as shown in Figure 2. When students are unwell or have abnormal body temperature measurement, they must first seek medical attention, and then choose hospital isolation, home isolation, home quarantine, or self-health management according to the doctor’s instructions and government regulations. The task team members established the campus protection measures according to the guidelines from Taiwan CDC with timely updates. The details were further customized by team members from each unit. All protection measures were adjusted through a rolling correction mechanism. Although a zero COVID-19 case record noted in campus cannot be entirely attributed to the performance of measures, we proposed that the established protection measures on campus might be useful and practical.

### 2.4. Practical Measures in Executing CPMCV-19

As shown in Table 1, the essential components in established measures were as follows: no close contact communication (team telecommunication by using mobile application LINETM, administrative teleconference using Microsoft Teams (Microsoft Co., Redmond, WA, USA), teleteaching simulation by TronClass platform (Wisdom Garden Co., New Taipei City, Taiwan)), daily campus sterilization on workdays via 0.05% (500 ppm) sodium hypochlorite (cleaning and disinfection of environmental surfaces in the context of COVID-19: Interim guidance, 15 May 2020, World Health Organization), school gate control with daily temperature measuring (cut-off 37.5 °C by frontal temperature), social distancing (1.5-m indoors and 1-m outdoors), activity restrictions (100 persons indoors and 500 persons outdoors), personal hygiene control (facemasks, handwashing education, desktop sneeze guards in classrooms and restaurants), and situational awareness (regulations and standard operating procedures, as well as execution analysis). Official announcements to students and parents were promptly posted on the school website. An example is listed in Appendix A.

## 3. Results

### 3.1. Establishment of Regulations, Standard Operating Procedures, and Decision-Making Algorithms

As shown in Table 2, when our school faced the outbreak of COVID-19, the epidemic prevention measures and actions taken, with a total of 21 regulations, standard operating procedures, and decision-making algorithms were established. After rolling corrections, the finalized decision-making algorithms are shown in Figure 2.

### 3.2. Effectiveness of CMPCV-19

During the 100 days of operation (from 26 January to 5 June 2020), the mean time spent for frontal temperature measuring was 2.7 ± 0.3 s per person. The mean on-duty time for individual personnel to control the gate and measure temperature was 3.5 h per day. In total, 31 persons with loss of taste/smell or fever were detected on campus and sent to the hospital for screening within 1 hour. A total of 6 persons were instructed to observe self-health management due to possible contact or travel history, and none were diagnosed with COVID-19 infection. Cancellation of 32 school activities with teleconferences held as alternatives successfully avoided physical contact for 14,400 person-times.

### 3.3. Cost and Time for CMPCV-19 Performance

A total budget of USD 27,100 was used for CMPCV-19 in this 100-day period. This budget was mainly used to purchase items needed for epidemic prevention including forehead temperature detectors, masks, alcohol, detergents, disinfectants, and the like. The administrative teleconferences using Microsoft Teams were held 3 times, for an accumulated time of 7 h in total. The teleteaching simulations by TronClass platform were performed twice, with a combined time of 6 h.

## 4. Discussion

The timely established campus protection measures for COVID-19 in an approximate 5000-person college were effective and practical. They can be used as reference for schools in a pandemic, such as COVID-19.

Since the 2003 outbreak of severe acute respiratory syndrome (SARS) in Taiwan, the medical and educational institutions have established various strategies for preventing outbreaks of infectious diseases and proposed response mechanisms for distributing resources efficiently during management operations [12,13]. The experience in responding to SARS with a 17-year interval aided the rapid response to cope with the COVID-19 pandemic in educational institutions. However, the viral characters of COVID-19 are distinct to SARS. For instance, the longer latency made campus protection from case clustering a vigorous challenge. Notably, a longer incubation time for SARS-CoV-2 infection with mean time 5.2 days (95% confidence interval (CI), 4.1 to 7.0) and the 95th percentile of the distribution at 12.5 days [14], indicated a need to adjust screening and control policies. It was suggested that potentially exposed subjects are required to be isolated for 14 days to avoid the risk of further transmission. As for symptoms/signs other than fever, loss of taste or loss of smell with severity greater than moderate were reported in 37.2% of patients positive for COVID-19 [15]. Taken together, we established the campus protection measures for COVID-19 by combining the experience from SARS and updating the information for COVID-19.

The major value of this work is to provide a timely reference for schools with similar scale to establish their own campus protection measures for the COVID-19 pandemic. The limitations of this work are that no actual positive COVID-19 cases were detected in the campus and no long-term follow-up data could be provided. The uniqueness of this work is the combination of global COVID-19 situation, government infectious disease control policy, and college campus environment to establish protection measures that are timely and practical for execution. To the best of our knowledge, no such timely and well-organized measures have yet been published about the COVID-19 pandemic. To assess the execution efficiency of the campus protection measures, the following must be considered: efficacy, time consumption, costs, and labor. The performance of CPMCV-19 resulted in avoidance of physical contact in a significant quantity and rapid processing for fever or symptom-detected individuals by spending reasonable costs of budget, time and labor, indicative of effective and practical measures for campus protection. These execution results not only provide physical protection but also psychological disburden for campus crisis and human stress [16,17,18].

## 5. Conclusions

The established campus protection measures for COVID-19 was practical and might be effective. However, it may need adjustments according to local culture and social characteristics. It may serve as a useful template for other colleges and small institutions that have limited resources.

## Figures and Tables

**Figure 1 healthcare-08-00308-f001:**
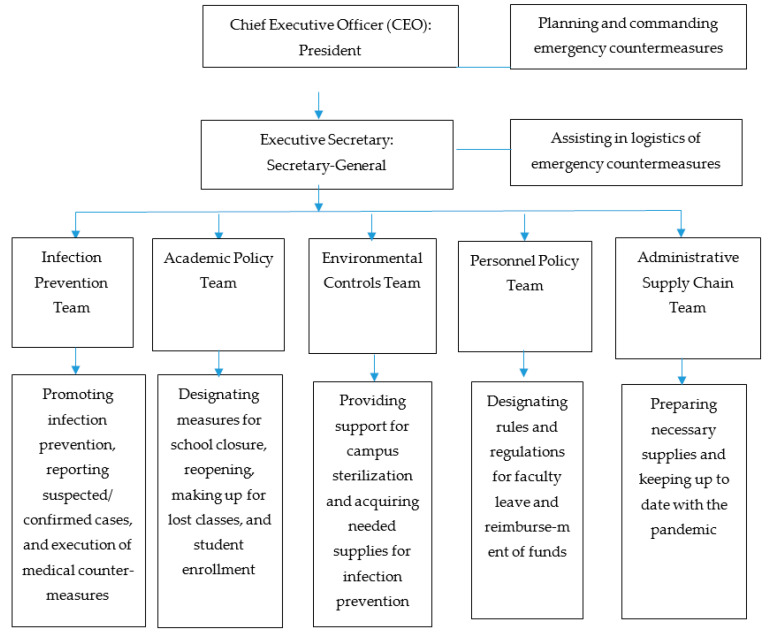
Organization of interdepartmental task force team in a college.

**Figure 2 healthcare-08-00308-f002:**
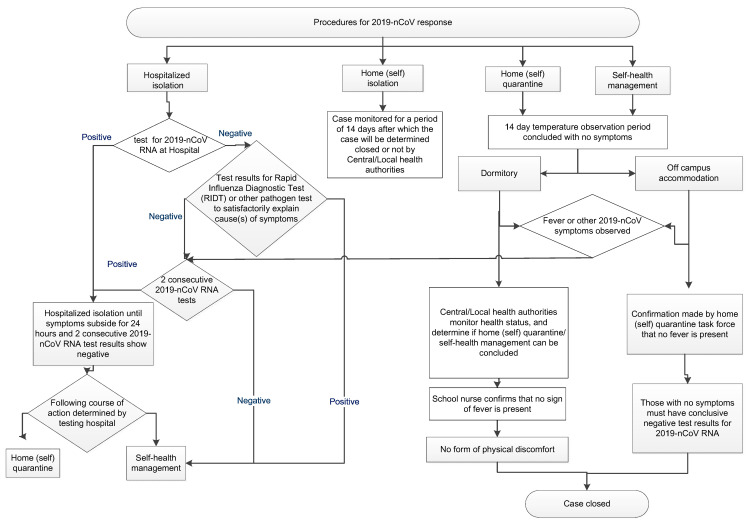
Decision-making algorithm flowchart for the coronavirus disease 2019 (COVID-19) pandemic in a college.

**Table 1 healthcare-08-00308-t001:** Essential components in established measures.

Essential Components	Established Measures
No close contact communication	1.Team telecommunication by using mobile application LINE 2. Administrative teleconference using Microsoft Teams 3. Teleteaching simulation on TronClass platform
Sterilization	Campus sterilization on school days
Temperature control	School gate control with daily temperature measuring < forehead 37.5 °C/inner ear 38 °C
Social distancing	1.5-m indoors and 1-m outdoors
Activity restrictions	<100 persons indoors and <500 persons outdoors
Personal hygiene control	Facemasks, handwashing education, desktop sneeze guards in classrooms and restaurants.
Situational awareness	Regulations and standard operating procedures, as well as execution analysis.

**Table 2 healthcare-08-00308-t002:** Regulations, precautions, and standard operating procedures (SOPs).

Item	Date Since First Confirmed Case	Description of Item	The Name of the Team
1	6	Establishment of school’s Infection Prevention Task Force	Infection Prevention Team
2	14	Establishment of school’s COVID-19 Infection Prevention Plan	Academic Policy Team
3	14	Letter published to all parents informing details of the COVID-19 Virus and school’s countermeasures	Infection Prevention Team
4	36	Establishment of COVID-19 protocols for school closure, reopening, and making up for lost classes	Academic Policy Team
5	37	Schedule set for periodical sterilization of campus grounds	Environmental Controls Team
6	45	Precaution notification of COVID-19 related false information and fake news	Administrative Supply Chain Team
7	50	SOP for students on campus with a fever	Administrative Supply Chain Team
8	50	SOP for ill dormitory boarders	Infection Prevention Team
9	50	SOP for dormitory boarders’ infection prevention	Infection Prevention Team
10	58	COVID-19 transition procedures for Taiwanese students studying abroad to repatriate and continue schooling	Infection Prevention Team
11	60	Faculty required to have temperature taken before boarding school bus	Infection Prevention Team
12	60	Establishment of dormitory boarders’ environment sterilization protocols and heath guidelines	Environmental Controls Team
13	64	Dormitory personnel infection prevention training and event simulation	Personnel Policy Team
14	67	Classroom infection prevention guidelines during COVID-19 pandemic	Academic Policy Team
15	70	Student self-moderation guidelines for off-campus accommodation	Infection Prevention Team
16	77	Faculty school bus infection prevention guidelines	Infection Prevention Team
17	86	Sterilization records for April, Guandu Campus	Environmental Controls Team
18	86	Sterilization records for April, Sanzhi Campus	Environmental Controls Team
19	96	Promotion of MOE’s School Dormitory Infection Prevention Guidelines	Infection Prevention Team
20	127	Sterilization records for May, Guandu Campus	Environmental Controls Team
21	127	Sterilization records for May, Sanzhi Campus	Environmental Controls Team

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
