# Peer review of "The First 100 Days: Establishment and Effectiveness of Campus Protection Measures at a College during the COVID-19 Pandemic"

_healthcare, 2020, doi:10.3390/healthcare8030308_

Round 1
Reviewer 1 Report
Thank you for allowing me to review this work. I hope you find my comments useful.
Abstract:
You need some context in the abstract. Otherwise it is very confusing to the reader. In which country is the project set? Is it a university or school? What is CPMCV-19?
What is frontal temperature measuring? What is being sterilized? What is the control gate?
Why do you discuss loss of taste and smell when you are measuring or screening temperature? I am unsure how the measures were effective. No-one was detected with COVID-19, therefore I am unsure how the program protected the school/university students.
Introduction:
Is the study set in Taiwan? Make this clear please? Again more context is needed. Why is the study/project important? Was it carried out in a school? how big? what age of students? how many students?
Materials and methods
Where is your ethical discussion?
How were the official measures customised and why? What does the 'rolling correction mechanism" entail?
Figure 2:
This needs more explanation. does the flow chart begin with a tested student via temperature checking at the school or university?How do you decide which of the 4 responses to choose (from hospitalisation to self-health management)? Is this flow chart based on government guidelines? is it adjusted? how? what happens if a student has COVID - to the college and its operations?
The second part of the method seems to be a list of measures. This needs explaining and interpreting. What is the source? were the measures useful?
I am unsure the term 'sterilization' is correct here. Do you mean sterilization of floors and surfaces by chemicals?
Results
I am unsure what table 2 refers to.
How were people with loss of taste and smell detected? The significance of this test needs to be discussed in the literature/ background section.
Discussion:
I am unsure how you can conclude the project was effective and practical. You have not provided evidence for either.
Your discussion does not flow from your findings. You introduce new issues here.
Can you discuss the value of this work? the rigour of the study? its limitations? Most importantly can you explain its uniqueness- what it adds to the literature on this topic? what is new here?
Your conclusion is more of a recommendation or acknowledgment of limitations.
Author Response
Response to Reviewers (healthcare-887019)
Thank you for your valuable comments and suggestions. The reviewers’ comments were very helpful for revising and improving our manuscript. To improve our manuscript, we have processed the English editing service for the text and attach a Certificate of English Editing for reference. Please find our point-by-point responses to your comments below.
Reviewer #1:
Abstract
You need some context in the abstract. Otherwise it is very confusing to the reader. In which country is the project set? Is it a university or school? What is CPMCV-19?
Response: Thank you for these valuable suggestions. As the reviewer rightly mentioned, essential details were missing. This project was set in Taiwan in a junior college called MacKay Junior College of Medicine, Nursing and Management. CPMCV-19 stands for campus protection measures for COVID-19. We have defined CPCMV-19 in the Abstract and text at its first presence.
What is frontal temperature measuring? What is being sterilized? What is the control gate?
Response: Frontal temperature measuring refers to face-to-face measurement of forehead temperature using a non-contact forehead thermometer. The campus and classrooms were sterilized. The control gate refers to the control of personnel entry at the school’s entrance.
Why do you discuss loss of taste and smell when you are measuring or screening temperature? I am unsure how the measures were effective. No-one was detected with COVID-19, therefore I am unsure how the program protected the school/university students.
Response: Fever is one of the primary symptoms of COVID-19, and loss of taste/smell has been officially listed as symptoms of COVID-19. Therefore, we measured the body temperature for screening for individuals with COVID-19 and interrogated all those who entered the premises on whether they had experienced loss of smell/ taste. Although no one was infected with COVID-19, this drill ensured that individuals with symptoms did not enter the site . It is our understanding that the application of campus protection measures such as those established for COVID-19 in the present study may need to be tailored per the regional culture and practices as well as social characteristics.
Introduction
Is the study set in Taiwan? Make this clear please? Again more context is needed. Why is the study/project important? Was it carried out in a school? how big? what age of students? how many students?
Response: Thank you for these essential comments. We apologize for not providing these details initially. This study was set in MacKay Junior College of Medicine, Nursing and Management, Taiwan. This school comprises approximately 4,200 students aged 16–20 years. This sentence was rewritten and corrected in the revised manuscript (page 2, lines 60–61). The findings of this study can be applied to the campus protection measures for COVID-19; however, they may need to be tailored per the regional culture and practices as well as social characteristics .
Materials and methods
Where is your ethical discussion?
How were the official measures customised and why? What does the 'rolling correction mechanism" entail?
Response: Thank you for these critical comments. All departments’ principal supervisors and selected secondary supervisors, 30 persons in total, were recruited and assigned to various task groups. We have obtained signed informed consent from all the 30 persons involved. The official measures were established by the 30 leaders who agreed to meet and discuss. The rolling correction mechanism implies that when the project is being implemented, it is revised as it is being implemented, and the project is adopted and revised at executive administration meetings.
Figure 2:
This needs more explanation. does the flow chart begin with a tested student via temperature checking at the school or university? How do you decide which of the 4 responses to choose (from hospital isolation to self-health management)? Is this flow chart based on government guidelines? is it adjusted? how? what happens if a student has COVID - to the college and its operations?
Response: Thank you for these critical comments. As the reviewer rightly mentioned, the flow chart begins with a student being examined via temperature monitoring at our school. When students are unwell or have an abnormal body temperature, they must first seek medical attention, following which they can choose hospital isolation, home isolation, home quarantine, or self-health management according to the doctor’s instructions and government regulations. This sentence was rewritten and corrected in the revised manuscript (page 2–3, lines 86–90).
The second part of the method seems to be a list of measures. This needs explaining and interpreting. What is the source? were the measures useful?
Response: As provided in the Materials and Methods, the task team members established the campus protection measures according to the guidelines from the Taiwan Centers for Disease Control (CDC) with timely updates. The details were further customized by team members from each unit. All protection measures were adjusted using the rolling correction mechanism. Although the zero COVID-19 cases noted on the campus cannot be entirely attributed to the implementation of these measures, we propose that the established protection measures on campus might be useful and practical.
I am unsure the term 'sterilization' is correct here. Do you mean sterilization of floors and surfaces by chemicals?
Response : Sterilization refers to the use of disinfectants to disinfect the campus environment and classroom floors and surfaces.
Results
I am unsure what table 2 refers to.
Response: Table 2 explains the epidemic prevention measures and actions adopted when our school faced the COVID-19 outbreak. This sentence was rewritten and corrected in the revised manuscript (page 5, lines 140–142).
How were people with loss of taste and smell detected? The significance of this test needs to be discussed in the literature/ background section.
Response: There was no specific test conducted for assessing the loss of taste/smell. This was simply based on the responses provided by individuals when being examined by a doctor.
Discussion:
I am unsure how you can conclude the project was effective and practical. You have not provided evidence for either.
Your discussion does not flow from your findings. You introduce new issues here.
Response: We agree that it is not pragmatic to claim that the project was effective and practical. Although the zero COVID-19 cases noted on the campus cannot be entirely attributed to the implementation of measures, we propose that the established protection measures on campus might be useful and practical. To address this issue more conservatively, the sentence “The established campus protection measures for COVID-19 in our college were effective and practical” in the Abstract and Discussion has been corrected to “The established campus protection measures for COVID-19 was practical and might be effective.” As for the content of the Discussion, we elaborated on the thought process and historical background, including the management experience of SARS in Taiwan as a basis for coronavirus, to further strengthen the establishment of the campus protection measures.
Can you discuss the value of this work? the rigour of the study? its limitations? Most importantly can you explain its uniqueness- what it adds to the literature on this topic? what is new here?
Response: The strength of this work is that we provide a timely reference for schools with a similar scale to establish their own campus protection measures for the COVID-19 pandemic. The limitations of this work include the following: no positive COVID-19 case was detected on campus and no long-term follow-up data could be provided. The uniqueness of this work is that the combination of global COVID-19 situation reports, government infectious disease control policies, and college campus environment data were used to establish timely and practical protection measures . To the best of our knowledge, no such timely and well-organized measures published during the COVID-19 pandemic exist.
Your conclusion is more of a recommendation or acknowledgment of limitations.
Response: Thank you for these comments. The conclusion was rewritten in the revised manuscript according to the reviewer’s suggestion (page 7, lines 197–199).
“The established campus protection measures for COVID-19 was practical and might be effective. However, it may need adjustments according to local culture and social characteristics. It may serve as a useful template for other colleges and small institutions that have limited resources.”
[A1]Dear Author,
Although this response letter did not require heavy or extensive revisions, I have carefully checked it for language, readability, clarity, and an appropriate tone. As the manuscript has not been submitted, please ensure consistency between the revised manuscript and the response letter. I have also crosschecked your responses and found that many of the responses accurately and adequately addressed the issues raised by the reviewers. I have flagged them to you in the appropriate comment boxes.
The section/page and line numbers where applicable texts should be updated. Kindly verify that these are in order. Should you make additional changes after your review, kindly update these.
Please carefully read the response letter and accept the changes I made if they align with your intention. If you find my questions in the response letter useful, you can also attend to them.
I wish you best of luck in the publication of this manuscript.
[A2]Please note that the reviewers’ comments have not been edited.
[A3]I have revised this to the boldfaced style to clearly distinguish it from the response.
[A4]Please indicate what does the red and blue fonts stand for.
[A5]I have added this sentence to answer to the final question. Please review.
[A6]Changes were made here to improve the clarity and readability of this part. Please check whether the revised part retains the intended meaning.
[A7]As the reviewer mentioned, the term sterilization is not suited here. I think the authors mean ‘disinfection.’ Please review.
[A8]Please mention what scale is being spoken about.
[A9]This sentence is not very clear. I have added ‘reports’ after situation and ‘data’ after environment as these phrases were incomplete. Please review.
[A10]I edited this for a more formal tone.
Reviewer 2 Report
Manuscript 887019
Title: Establishment and Effectiveness of Campus Protective Measures in a College during the COVID-19 Pandemic.
This paper reports on steps taken to control the spread of COVID-19 in a college campus. It reports on effectiveness and costs for measures taken to protect the college community. The measures taken include regulation development, standard operating procedures, and decision-making algorithms. Seven essential components were selected as primary drivers: 1) no close contact communication; 2) sterilization; 3) temperature control; 4) social distancing; 5) activity restrictions; 6) personal hygiene control ; and 7) situational awareness.
Major Concerns
- Frame the work around the first 100 days of the outbreak. Consider changing the title to: The First 100 Days: Campus Protection Measures in a College during the COVID-19 Pandemic.
- What are the measures of effectiveness? The paper seems to be limited to process measures and does not really address outcomes. Costs should be provided in more detail. Consider explicitly framing the focus on outcomes described as: budget, time, and labor.
- I question the value of Figure 2 for the paper. It should be deleted. There are things in the Figure that are not discussed in the paper and the figure raises more questions than it answers. For example, what is the “home (self) quarantine task force”? The last statement on page 2 (lines 75-77) makes it sound like Figure 2 will present a decision algorithm for closing the college for 14 days. This does not seem to be one of the decision nodes on the Figure.
- Page 6, Lines 140-143 . The costs need more detail. It appears that 7 hours of Microsoft Teams and 6 hours of tele-teaching cost $27,100.
- Page 6. Conclusion is weak. Consider revising to state that this study describes the approach taken to address the COVID 19 pandemic from the perspective of a college. It may serve as a useful template for other colleges and small institutions that have limited resources.
Minor Concerns
- Drop the last sentence on page 1 (lines 42-44). It is repeated in the next paragraph.
- It would be helpful to have more informative short labels for each of the 5 teams in Figure 1. Maybe something like: Infection Prevention Team; Academic Policy Team; Environmental Controls Team; Personnel Policy Team; Administrative Supply Chain Team.
- In Table 2, add the name of the Team from Figure 1 that took the lead or had primary responsibility for attaining each milestone on the list.
- Page 6, Line 160. “camps” should be “campus”.
Author Response
Response to Reviewers (healthcare-887019)
Thank you for your valuable comments and suggestions. The reviewers’ comments were very helpful for revising and improving our manuscript. To improve our manuscript, we have processed the English editing service for the text and attach a Certificate of English Editing for reference. Please find our point-by-point responses to your comments below.
[A1]Dear Author,
Although this response letter did not require heavy or extensive revisions, I have carefully checked it for language, readability, clarity, and an appropriate tone. As the manuscript has not been submitted, please ensure consistency between the revised manuscript and the response letter. I have also crosschecked your responses and found that many of the responses accurately and adequately addressed the issues raised by the reviewers. I have flagged them to you in the appropriate comment boxes.
The section/page and line numbers where applicable texts should be updated. Kindly verify that these are in order. Should you make additional changes after your review, kindly update these.
Please carefully read the response letter and accept the changes I made if they align with your intention. If you find my questions in the response letter useful, you can also attend to them.
I wish you best of luck in the publication of this manuscript.
Reviewer #2:
Major Concerns
- Frame the work around the first 100 days of the outbreak. Consider changing the title to: The First 100 Days: Campus Protection Measures in a College during the COVID-19 Pandemic.Revised title:
- Response: Thank you for these comments. We have modified the title in the revised manuscript according to the reviewer’s suggestion.
“The First 100 Days: Campus Protection Measures in a College during the COVID-19 Pandemic.”
- What are the measures of effectiveness? The paper seems to be limited to process measures and does not really address outcomes. Costs should be provided in more detail. Consider explicitly framing the focus on outcomes described as: budget, time, and labor.Revised text:
- Response: This sentence was rewritten and corrected in the revised manuscript (page 6, lines 154–159).
“This budget was mainly used to purchase items needed for epidemic prevention including forehead temperature detectors, masks, alcohol, detergents, disinfectants, and the like. The administrative teleconferences using Microsoft Teams were held 3 times for an accumulated time of 7 hours in total. The teleteaching simulations by TronClass platform were performed twice with a combined time of 6 hours.”
- I question the value of Figure 2 for the paper. It should be deleted. There are things in the Figure that are not discussed in the paper and the figure raises more questions than it answers. For example, what is the “home (self) quarantine task force”? The last statement on page 2 (lines 75-77) makes it sound like Figure 2 will present a decision algorithm for closing the college for 14 days. This does not seem to be one of the decision nodes on the Figure.Revised text:
- “When students are unwell or have abnormal body temperature measurement, they must first seek medical attention, and then choose hospital isolation, home isolation, home quarantine, or self-health management according to the doctor's instructions and government regulations. The task team members established the campus protection measures according to the guidelines from Taiwan CDC with timely updates.”
- Response: This sentence was rewritten and corrected in the revised manuscript (page 2–3, lines 85–89 ).
- Page 6, Lines 140-143 . The costs need more detail. It appears that 7 hours of Microsoft Teams and 6 hours of tele-teaching cost $27,100.
- Response: This budget was mainly used to purchase items needed for epidemic prevention including forehead temperature detectors, masks, alcohol cleaning agents, detergents, disinfectants, etc. This sentence was rewritten and corrected in the revised manuscript (page 6, lines 154–156).
- Page 6. Conclusion is weak. Consider revising to state that this study describes the approach taken to address the COVID 19 pandemic from the perspective of a college. It may serve as a useful template for other colleges and small institutions that have limited resources.Revised text:
- Minor Concerns
- “It may serve as a useful template for other colleges and small institutions that have limited resources.”
- Response: Thank you for these comments. The conclusion was rewritten in the revised manuscript according to the reviewer’s suggestion (page 7, lines 198–199).
- Drop the last sentence on page 1 (lines 42-44). It is repeated in the next paragraph.
- Response: Thank you for the comment. The sentence has been removed in the revised manuscript according to the reviewer’s suggestion.
- It would be helpful to have more informative short labels for each of the 5 teams in Figure 1. Maybe something like: Infection Prevention Team; Academic Policy Team; Environmental Controls Team; Personnel Policy Team; Administrative Supply Chain Team.
- Response: We have revised as per the reviewer’s suggestion. Please refer to the 5 teams in Figure 1 of the revised manuscript.
- In Table 2, add the name of the Team from Figure 1 that took the lead or had primary responsibility for attaining each milestone on the list.
- Response: We have revised as per the reviewer’s suggestion. Please refer to Table 2 of the revised manuscript.
- Page 6, Line 160. “camps” should be “campus”.
- Response: This word was rewritten and corrected in the revised manuscript (page 7, line 192).
[A1]Please note that these details are similar to those of the previous response.
[A2]This acronym was already introduced in the text and may be consistently used without having to state its full meaning.